# The hepatic transcriptome is differentially regulated by a standardized meal in healthy individuals compared to patients with fatty liver disease

Josephine Grandt[1,2‡], Christian D. Johansen[2‡], Anne-Sofie H. Jensen[1,2], Mikkel P. Werge[1], Elias B. Rashu[1], Andreas Møller[1], Anders E. Junker[1], Lise Hobolth[1], Christian Mortensen[1], Mogens Vyberg[3,4], Reza Rafiolsadat Serizawa[3], Søren Møller[5,6], Lise Lotte Gluud [1,6‡], Nicolai J. Wewer Albrechtsen [2,6,7,8‡]*

**1** Gastro Unit, Copenhagen University Hospital Hvidovre, Denmark, **2** Novo Nordisk Foundation Center for Protein Research, Faculty of Health and Medical Sciences, University of Copenhagen, Copenhagen, Denmark, **3** Department of Pathology, Copenhagen University Hospital Hvidovre, Denmark, **4** Center for RNA Medicine, Department of Clinical Medicine, Aalborg University, Copenhagen, Denmark, **5** Department of Clinical Physiology and Nuclear Medicine, Center for Functional and Diagnostic Imaging and Research, Copenhagen University Hospital Hvidovre, Denmark, **6** Department of Clinical Medicine, Faculty of Health and Medical Sciences, University of Copenhagen, Copenhagen, Denmark, **7** Department of Clinical Biochemistry, University Hospital Copenhagen - Bispebjerg and Frederiksberg, Copenhagen, Denmark, **8** Copenhagen Center for Translational Research, University Hospital Copenhagen - Bispebjerg and Frederiksberg, Copenhagen, Denmark

‡ These authors share first authorship on this work.
* Nicolai.Albrechtsen@regionh.dk

## Abstract

The human liver is dynamic organ with minute to hourly adaptions in response to feeding. Patients with non-alcoholic fatty liver disease (NAFLD) and cirrhosis have altered transcriptomic features compared to controls but how and if food intake affects such is unknown in humans. Our aim was to investigate the hepatic transcriptome at both fasting and postprandial states in patients with NAFLD, cirrhosis, and healthy controls and secondly to develop a browsable resource enabling easy and unrestricted access to such data. We hypothesized that hepatic transcriptome differed between groups, and this was also regulated by food intake.

We obtained liver tissue by transjugular liver biopsies from patients with NAFLD (n = 9, mean age 49 (16 SD) y, BMI 35 (5) kg/m$^2$), cirrhosis (n = 9, age 61 (11) y, BMI 32 (5) kg/m$^2$) and healthy controls (n = 10, age 25 (3) y, BMI 23 (3) kg/m$^2$). The hepatic transcriptome was sequenced using NGS and evaluated in bioinformatic analyses to assess differentially expressed genes (DEG) and gene ontology biological processes (GOBP). We identified 553 DEG between healthy controls and patients with NAFLD, 5527 DEG between healthy controls and patients with cirrhosis, and 3898 DEG in NAFLD compared with cirrhosis. A hitherto uncharacterized gene (MET proto-oncogene) was differentially expressed in human NAFLD and cirrhosis. The hepatic

**Data availability statement:** The code is available on GitHub https://github.com/nicwin98). All raw data are available freely through the EBL server using the accession number: E-MTAB-12807.

**Funding:** The study and Josephine Grandt, Christian D. Johansen, Anne-Sofie H. Jensen, and Nicolai J. Wewer Albrechtsen were supported by Novo Nordisk Foundation Excellence Emerging Investigator Grant – Endocrinology and Metabolism (Application No. NNF19OC0055001), European Foundation for the Study of Diabetes Future Leader Award (NNF21SA0072746) and Independent Research Fund Denmark, Sapere Aude (1052-00003B). Novo Nordisk Foundation Center for Protein Research is supported financially by the Novo Nordisk Foundation (Grant agreement NNF14CC0001). The funders had no role in study design, data collection and analysis, decision to publish, or preparation of the manuscript.

**Competing interests:** The authors have declared that no competing interests exist.

transcriptome changed significantly during a standardized meal and these changes were blunted in patients with NAFLD and cirrhosis. GOBP analyses revealed an increase in pro-inflammatory and pro-fibrotic genes in NAFLD and cirrhosis, as well as a decrease in genes related to metabolism. Data were made browsable using two web-based apps. The hepatic transcriptome is differentially regulated by a standardized meal in healthy individuals compared to patients with fatty liver disease.

## 1. Introduction

Non-alcoholic fatty liver disease (NAFLD) is the most common liver disease worldwide, with an estimated 25% prevalence, and is becoming the main cause for liver transplantations [1]. NAFLD encompasses simple steatosis, also referred to as non-alcoholic fatty liver (NAFL), which can progress to non-alcoholic steatohepatitis (NASH) that may lead to fibrosis development and, eventually, even cirrhosis. The pathophysiological mechanisms of NAFLD development and progression are still not clear, but multiple factors, including insulin resistance and genetic factors, have been identified [2]. Further insights into the pathophysiology of NAFLD and its progression to cirrhosis are crucial to identifying non-invasive markers and novel drug targets.

RNA sequencing methods have progressed over the last decade. Until recently, transcriptomics studies used micro-array platforms with preselected genes to study differences in gene expression. Next-generation sequencing (NGS) provides a hypothesis-free approach with high specificity and sensitivity allowing insights into molecular mechanisms by identifying crucial genes and functional pathways. Previous studies have evaluated the hepatic transcriptome focusing on the progression from NAFL to NASH [3–5]. Hepatic RNA-seq datasets from patients with NAFLD, cirrhosis, and healthy controls are to our knowledge not been publicly available. Likewise, the differential expression of genes in the fasting and postprandial state remains unknown. The latter may be of importance as the liver has a vital role in postprandial metabolism and the adaptive response to a meal may be blunted in patients with NAFLD and cirrhosis thereby contributing to numerous metabolic complications such as diabetes and cardiovascular diseases [6].

Here, we aimed to 1) investigate gene expression differences of the liver in patients with NAFLD, cirrhosis, and healthy controls; 2) explore if the human liver transcriptome differs between the fasting and fed state in the three groups, and 3) develop an R based, browsable resource that makes data explorable for everyone through a web-browser (https://weweralbrechtsenlab.shinyapps.io/PLS_Groups/ & https://weweralbrechtsenlab.shinyapps.io/PostprandialLiver/), and all raw data and coding deposited publicly at GitHub(https://github.com/nicwin98) for researchers to use. We hypothesized that hepatic transcriptome differed between groups, and this was also regulated by food intake.

The data provided may provide novel insight into human physiology and how liver diseases are affecting the body ability to handle nutrients. The latter may enable discoveries into molecular drivers of the increased risk of metabolic disease such as diabetes in patients with chronic liver diseases.

## 2. Materials and methods

### 2.1. Study design and ethics

The study included 28 participants from the postprandial human liver study (Supplementary Figure 1). In brief, the post-prandial human liver study included 9 patients with NAFLD, 9 patients with cirrhosis, and 10 healthy controls. Initially, 29 participants were randomized to undergo a liver biopsy in the fasting state or 60 min after a standardized meal. Blood samples were collected at several time points. One of the participants with cirrhosis did not undergo a liver biopsy (a biopsy was not possible for technical reasons). This study includes all remaining 28 participants with RNA sequencing biopsy specimens. The study design and patient characteristics has previously been reported [7]. Study participants were recruited between 2 March 2019 and 1 January 2022.

Participants were recruited at the outpatient clinic of the Gastro Unit at Hvidovre Hospital. All participants gave their written and oral informed consent to participate in the study. Baseline blood samples and anthropometric data were collected on the day of screening. Healthy controls were normal weight, had no chronic diseases or medication intake. There were no requirements regarding the aetiology of cirrhosis, Child-Pugh score, medication, or previous decompensation. Patients with ongoing alcohol abuse or malignant diseases were excluded

The local ethics committee (The Scientific Ethics Committees for the Capital Region) approved the study (H-18052725), and the study is registered as NCT03849235 at clinicaltrials.gov. The study was performed in compliance with the ethical guidelines of the Declaration of Helsinki.

### 2.2. Collection of transjugular liver biopsies

All participants underwent a transjugular liver biopsy with at least two passes. Part of the liver tissue was used for histological assessment. Two expert pathologists independently evaluated biopsies. The NAFLD severity was assessed using the NAFLD activity score (NAS score). The remaining tissue for transcriptomic analyses was placed in RNA later and stored at 4°C until later RNA sequencing analysis.

### 2.3. RNA sequencing by Illumina

RNA sequencing was performed using short-read Illumina sequencing technology as follows: RNA extraction and qualification were done using an AllPrep DNA/RNA Minikit (cat 80204- QIAgen) and 2100 Bioanalyzer instrument (Agilent Genomics), followed by treatment with the RNeasy Minikit (cat 74106 - QIAgen) and R-nase Free DNase set (cat 79254 - QIAgen) according to the manufacturer's protocol.

TruSeq Stranded Total RNA Library Prep Gold (cat. 20020599 – Illumina) and IDT for Illumina-TruSeq RNA UD Indexes (cat 20022371 – Illumina) were used for library preparation. Illumina Novaseq 6000 instrument was used for paired-end (2x 150 bp) sequencing of RNA-Seq libraries, and raw sequencing data were processed by bcl2fastq software version 2.20.0 (Illumina).

### 2.4. Raw data processing

FastQC 0.11.9 was used for quality control of fastq files, and Salmon (Version 1.4.0) was used to map reads to the human transcriptome (GRCh38.p13) and perform alignment. The quantification files were imported into R using the *tximeta* package, followed by summarization and prefiltering.

### 2.5. Data analysis

For explorative data analysis, the algorithm variance stabilizing transformation (VST) was used to normalize the data. Tendencies in the data were assessed by hierarchical clustering and a principal component analysis (PCA). We performed PCA analysis to visualize global gene expression patterns of study participants. *DESeq2* package (version 1.32.0) was used for differential expression analysis and correction for multiple testing was applied using the false discovery rate (FDR) with

an alpha of 0.05. For the differential expression analyses, we corrected for sequencing and library preparation biases in the regression model using *DESeq2* (version 1.32.0). Results were visualized in a volcano plot with the *Enhanced Volcano* package (version 1.10.0). Gene Ontology (GO) analysis was performed with the *clusterProfiler package* (version 4.0.0).

## 2.6. Browsable application, and data and code availability

The browsable web application was coded in *R* (version 4.2.0). The packages *Shiny* (version 1.7.2), *Enhanced Volcano* (version 1.10.0), and *DT* (version 0.24) were used to create the web application. The apps are available through https://weweralbrechtsenlab.shinyapps.io/PLS_Groups/ and https://weweralbrechtsenlab.shinyapps.io/PostprandialLiver/. The code is available on GitHub https://github.com/nicwin98). All raw data are available freely through the EBL server using the accession number: E-MTAB-12807.

## 3. Results

### 3.1. Patient characteristics

Participants with cirrhosis were older participants in the NAFLD group and healthy controls (Table 1). The patient characteristics have previously been reported [7]. Participants with NAFLD had a higher BMI than the two remaining groups. Histological scores are shown in Table 2.

### 3.2. Browsable app

We developed a browsable web application based on *R statistical software* to make our data easily accessible (*Shiny R* application). See https://weweralbrechtsenlab.shinyapps.io/PLS_Groups/ (group differences between healthy, NAFLD, and cirrhosis) and https://weweralbrechtsenlab.shinyapps.io/PostprandialLiver/ (postprandial versus fasting).

**Table 1. General participant characteristics.**

| Variable | Healthy | NAFLD | Cirrhosis |
|---|---|---|---|
| n | 10 | 9 | 9 |
| Age, years | 25 (3) | 49 (16) | 61 (11) |
| Male/ female | 5/5 | 4/5 | 5/4 |
| BMI, kg/m$^2$ | 23 (2) | 35 (5) | 32 (5) |
| Waist circumference, cm | 80 (9) | 113 (12) | 111 (11) |
| VAT, L | 0.8 (0.9) | 4.5 (1.9) | 4.1 (2.1) |
| T2D, n | 0 | 0 | 2 |
| A1c, mmol/mol | 33 (3) | 34 (7) | 36 (12) |
| Fasting glucose, mmol/L | 5.0 (0.3) | 5.5 (0.6) | 7.3 (2.6) |
| HOMA-IR | 1.4 (0.7) | 3.7 (2.0) | 8.3 (4.9) |
| Glucagon, pM | 6 (2) | 9 (6) | 15 (16) |
| Triglycerides, mmol/L | 0.8 (0.2) | 1.5 (0.6) | 1.5 (0.7) |
| HDL, mmol/L | 1.6 (0.2) | 1.4 (0.5) | 1.2 (0.5) |
| LDL, mmol/L | 1.9 (0.5) | 3.3 (0.6) | 2.3 (0.7) |
| ALT, U/L | 10 (5) | 31 (27) | 16 (10) |
| AST, U/L | 21 (4) | 40 (19) | 78 (85) |
| FIB-4 | 0.6 (0.2) | 1.0 (0.7) | 6.7 (3.6) |
| NAS score | 0 (0) | 2 (1) | 3 (2) |

Data is presented as mean (SD). BMI; Body Mass Index; VAT: Visceral Adipose Tissue; T2D: Type 2 Diabetes; A1c: Hemoglobin A1C; HOMA-IR: Homeostatic Model Assessment for Insulin Resistance; HDL: High-Density Lipoprotein; ALT: Alanine Transaminase, AST: Aspartate Transaminase; FIB-4: Fibrosis-4; NAS Score: NAFLD Activity Score.

**Table 2. NAFLD activity score in healthy controls, patients with NAFLD, and patients with cirrhosis.**

| Group | n | Steatosis | | | | Inflammation | | | Ballooning | | | Fibrosis | | | | |
|---|---|---|---|---|---|---|---|---|---|---|---|---|---|---|---|---|
| | | 0 | 1 | 2 | 3 | 0 | 1 | 2 | 0 | 1 | 2 | 0 | 1 | 2 | 3 | 4 |
| Healthy | 10 | 10 | – | – | – | 9 | 1 | – | 10 | – | – | 10 | – | – | – | – |
| NAFLD | 9 | – | 6 | 3 | – | 4 | 5 | – | 5 | 4 | – | 7 | – | 1 | 1 | – |
| Cirrhosis | 9 | – | 3 | 4 | 1 | 1 | 6 | 1 | 3 | 4 | 1 | – | – | – | 1 | 7 |

The application is an intuitive, interactive tool that allows users to create customized expression profiles for genes and the whole data set. Gene lists are accessible with log2FCs, p-values and adjusted p-values, and other statistical parameters. Users can visualize the data through interactive volcano plots, create, and export boxplots of DEG and customized dot plots of gene ontology biological processes, among others.

### 3.3. Principal component analysis (PCA)

To evaluate the global differences between groups we performed principal component analysis (PCA) on the dataset. The first PCA that accounted for the most variability in the three study groups showed that the transcriptome profiles of patients with cirrhosis separated clearly from patients with NAFLD and healthy controls (Fig 1A) as expected. Important differences were also identified when comparing patients with NAFLD and healthy controls (Fig 1B). No differences were observed between participants randomized to fasted versus a standardised meal likely due to the fact that the components visualized mainly reflect differences between groups.

### 3.4. Global transcriptional profile

Our global gene expression analysis revealed a total of 553 (2.5% of the gene pool) DEG when comparing NAFLD with healthy controls (256 upregulated and 297 downregulated in NAFLD) (Fig 2A and Supplementary Table 1 and Table 2). When comparing cirrhosis with NAFLD, we found 3898 (17%) DEG (2091 upregulated and 1807 downregulated in cirrhosis) and 5527 (24%) differentially expressed genes comparing cirrhosis with healthy controls. (2980 upregulated and 2547 downregulated in cirrhosis) (Fig 2B and C).

### 3.5. Differentially expressed genes in fasting versus postprandial conditions

In healthy controls, we identified 35 DEG in participants allocated to fasting versus those allocated to a standardised meal. Fewer DEG were identified when comparing the two groups among in participants with NAFLD or cirrhosis. One example of such differences between healthy and patients with cirrhosis is ANGPTL8, a protein known to increase in the circulation during a meal and that contribute to postprandial lipid metabolism. Another example is IGF1 which was differentially regulated in healthy individuals (downregulated postprandially) compared to patients with NAFLD (upregulated postprandially). 5 identical genes (NR0B2, SERTAD3, MIDN, IGF1, NKD1, Table 3) were found regulated by the meal in both healthy and NAFLD, whereas no overlap was observed for cirrhosis compared to NAFLD or controls.

When analysing DEG in all participants (adjusting for fasting or postprandial state), we found several differences between participants with NAFLD compared with healthy controls. Among the 10 most upregulated genes in NAFLD were genes related to insulin resistance (PRKCE), glucose metabolism and steatosis (PLIN1), glucose uptake regulation (ISM1), oxidative cell stress response (TP53INP1), and a proinflammatory marker (SERPINE 1) (Supplementary table 1).

In addition, our analyses showed upregulation of a novel obesity gene (TBX15) and a gene recently found associated to hepatic inflammation and apoptosis in mice (ZBTB33).

The 10 most significantly downregulated genes in NAFLD included genes protecting against steatosis (MET) and fibrosis (GAS5), cancer suppression (RBMS1), and genes downregulated in HCC (FNDC3B) (Supplementary table 1).

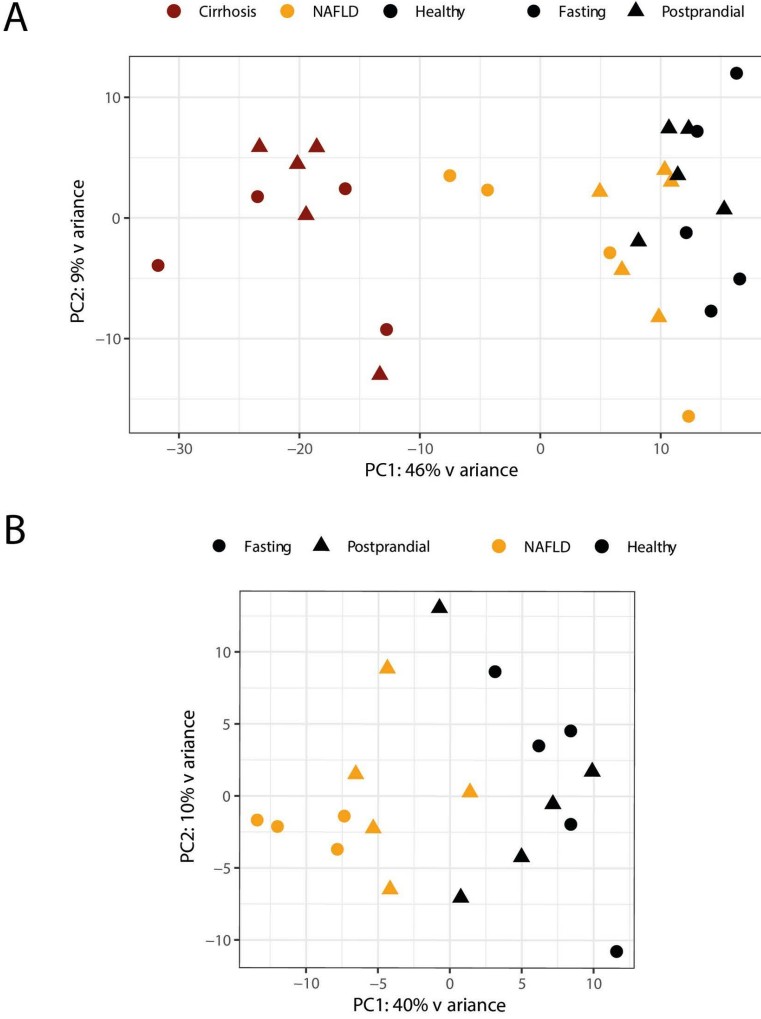

**Fig 1. Principal Component Analysis (PCA) separates patients with chronic liver disease from healthy controls independent of postprandial status.** (A) Including all participants (cirrhosis, NAFLD, and healthy controls). (B) Including participants with NAFLD and healthy controls. Triangles reflect postprandial condition and circle represent fasting conditions. Black circles and triangles represent healthy individuals, yellow patients with NAFLD, and red patients with cirrhosis.

For cirrhosis compared with NAFLD, the most significantly upregulated genes were related to liver cancer (ITGA2, GOLM1) [8,9] and inflammation (ITGA2) [10], inflammasome activation (JAG1) [10], and fibrosis (JAG1) [11,12]. One gene which is previously found to be associated with NAFLD progression (RGS5) [13], was downregulated in NAFLD. NCAM2 was differentially expressed in patients with NAFLD and cirrhosis compared to healthy individuals (Table 4).

The upregulated genes between cirrhosis and healthy included GSN, COL4A4, CHI3L1, and GLIS2. Among the most significant downregulated genes, we found POSTN, PACSIN3, and DNASE1L3.

Differential expression of COL4A4 and PACSIN3 was found in patients with NAFLD and cirrhosis compared to healthy individuals (see Table 4), and GSN, that is associated to HCC, was the most upregulated gene in cirrhosis. Interestingly, Periostin (POSTN) was both downregulated for cirrhosis compared to NAFLD and cirrhosis compared to healthy, although its overexpression was associated with inflammation, fibrosis, and HCC development, as well as metabolic disorders [14].

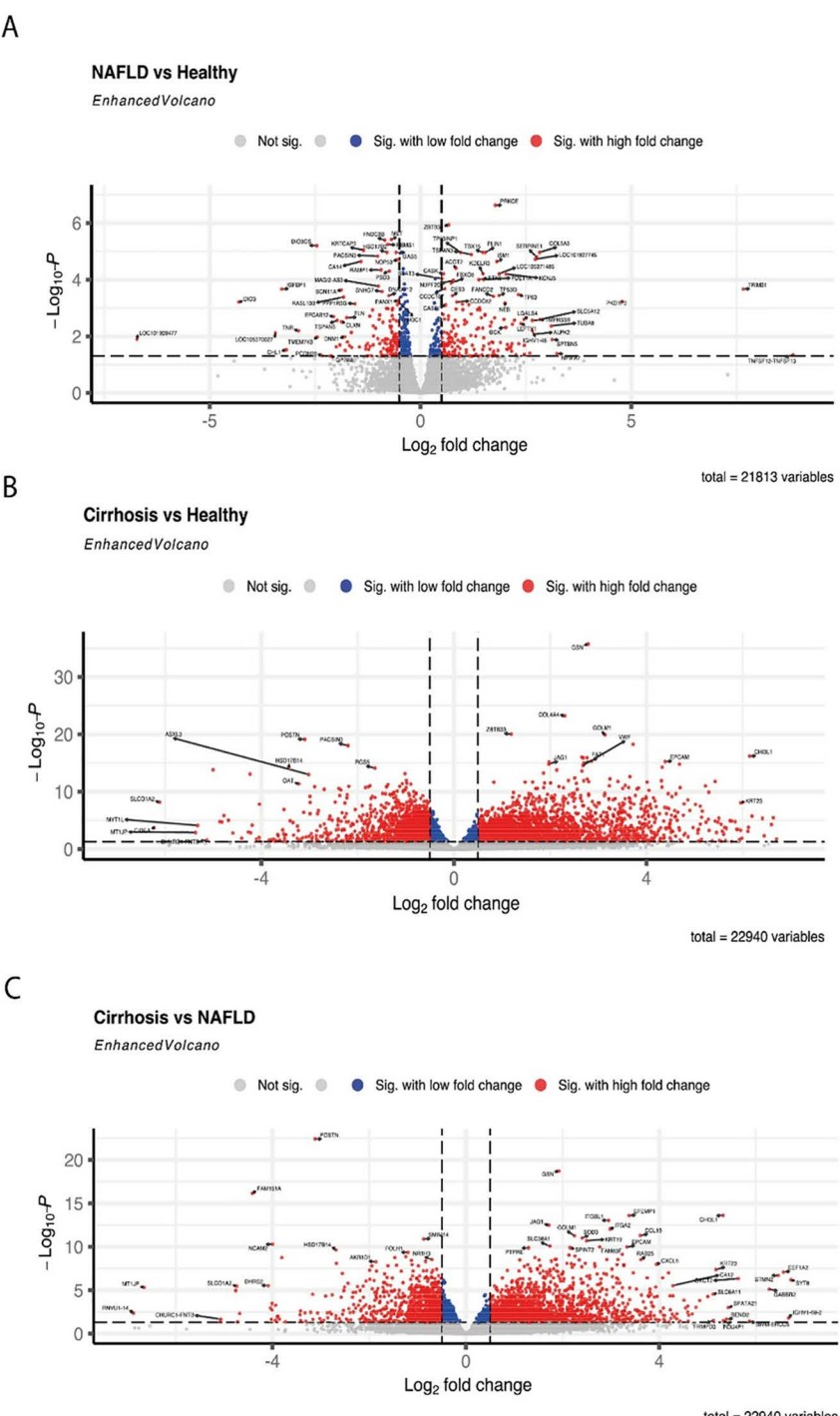

**Fig 2. Volcano plots showing the overall impact of group differences on the hepatic transcriptome.** In each panel, a XY plot is shown. The x-axis represents log2fold change between groups. The Y-axis the level of association (-log10 P-value) including correction for multiple testing. Each dot shown in the plots represent one transcript (gene). Those shown in blue are significant but with low fold change whereas those shown in red are with high fold change. For some of the genes, we have annotated names on graph for selected genes. Those shown in grey are not significantly different between groups. In the lower right corner, the total amount of genes included in the analysis is shown. (A) Differences between NAFLD and healthy. (B) Differences between Cirrhosis and healthy. (C) Differences between cirrhosis and NAFLD.

**Table 3. Differentially expressed genes (DEG) in healthy controls and participants with NAFLD allocated to a standardised meal (postprandial) vs fasting.** A fold change <1 reflect that the gene is downregulated in the postprandial state and a fold change of >1 reflect that the gene is upregulated in the postprandial state.

| Gene symbol | Gene name | Healthy postprandial vs fasting | | NAFLD postprandial vs fasting | |
|---|---|---|---|---|---|
| | | fold Change | adjusted p-value | fold Change | adjusted p-value |
| IGF1 | insulin like growth factor 1 | 0.48 | 0.0007 | 2.73 | 1.56E-05 |
| NR0B2 | nuclear receptor subfamily 0 group B member 2 | 2.64 | 0.0009 | 2.43 | 0.033 |
| SERTAD3 | SERTA domain containing 3 | 2.05 | 0.002 | 2.07 | 0.016 |
| NKD1 | NKD inhibitor of WNT signaling pathway 1 | 0.33 | 0.033 | 0.28 | 0.038 |
| MIDN | Midnolin | 1.61 | 0.044 | 1.88 | 0.008 |

**Table 4. The 20 most significant differentially expressed genes between groups.**

| Gene symbol | Gene name | NAFLD vs healthy | | Cirrhosis vs NAFLD | | Cirrhosis vs healthy | |
|---|---|---|---|---|---|---|---|
| | | p adj. | fold change | p adj. | fold change | p adj. | fold change |
| ZBTB33 | zinc finger and BTB domain containing 33 | 1.16E-6 | 1.58 | 5.82E-05 | 1.43 | 8.84E-21 | 2.27 |
| KDELR3 | KDEL endoplasmic reticulum protein retention receptor 3 | 6.19E-05 | 2.79 | .019 | 1.90 | 3.52E-10 | 5.25 |
| STAT3 | signal transducer and activator of transcription 3 | 9.23E-05 | 1.43 | .007 | 1.25 | 2.11E-12 | 1.77 |
| COL4A4 | collagen type IV alpha 4 chain | 0.00073 | 1.98 | 1.14E-08 | 2.42 | 5.72E-24 | 4.89 |
| NCAM2 | neural cell adhesion molecule 2 | 0.001 | 2.55 | 4.95E-11 | 0.06 | 4.18E-05 | 0.15 |
| ZNF28 | zinc finger protein 28 | 0.002 | 2.29 | 0.031 | 1.62 | 2.87E-09 | 3.53 |
| ABCA2 | ATP binding cassette subfamily A member 2 | 0.002 | 1.58 | 0.029 | 1.33 | 9.78E-09 | 2.02 |
| ZMAT3 | zinc finger matrin-type 3 | 0.004 | 2.06 | 0.001 | 1.84 | 2.99E-12 | 3.67 |
| TYMS | thymidylate synthetase | 0.004 | 4.52 | 0.033 | 2.22 | 2.70E-11 | 11.35 |
| JUNB | JunB proto-oncogene, AP-1 transcription factor subunit | 0.005 | 1.71 | 0.003 | 1.76 | 2.14E-08 | 2.82 |
| MET | MET proto-oncogene, receptor tyrosine kinase | 3.80E-06 | 0.62 | 0.009 | 0.75 | 4.18E-12 | 0.47 |
| PACSIN3 | protein kinase C and casein kinase substrate in neurons 3 | 1.46E-05 | 0.49 | 4.39E-06 | 0.45 | 9.05E-19 | 0.22 |
| CA14 | carbonic anhydrase 14 | 2.27E-05 | 0.38 | 0.002 | 0.46 | 2.72E-11 | 0.19 |
| PSD3 | pleckstrin and Sec7 domain containing 3 | 4.94E-05 | 0.60 | 0.006 | 0.68 | 1.31E-10 | 0.42 |
| MAGI2-AS3 | MAGI2 antisense RNA 3 | 0.00017 | 0.50 | 0.014 | 0.60 | 1.45E-09 | 0.30 |
| EEF1D | eukaryotic translation elongation factor 1 delta | 0.00034 | 0.78 | 0.012 | 0.83 | 4.74E-10 | 0.635 |
| DNAJC12 | DnaJ heat shock protein family (Hsp40) member C12 | 0.00036 | 0.59 | 0.00034 | 0.50 | 1.93E-11 | 0.28 |
| NR3C1 | nuclear receptor subfamily 3 group C member 1 | 0.00062 | 0.70 | 0.003 | 0.67 | 1.56E-08 | 0.48 |
| EIF3E | eukaryotic translation initiation factor 3 subunit E | 0.00076 | 0.75 | 0.017 | 0.82 | 1.44E-09 | 0.62 |
| SOX5 | SRY-box transcription factor 5 | 0.0018 | 0.51 | 5.14E-05 | 0.41 | 1.17E-11 | 0.22 |

An upregulation of GLIS2 has been associated to NASH progression, and downregulation of DNASE1L3 is found in HCC and correlates to poor prognosis [15,16].

Another gene that was significantly upregulated in cirrhosis compared to healthy controls and NAFLD patients was CHI3L1, which was previously found to be increased in inflammation, promoting angiogenesis, and may have diagnostic value for liver fibrosis [17].

### 3.6. Genes differentially expressed across all groups

A total of 111 DEG were identified when comparing all three groups of participants. Among the 10 most significantly upregulated and downregulated genes (Table 4) one gene (MET) that has been linked with NASH was clearly downregulated.

MET codes for a receptor which is a transmembrane protein; its ligand is hepatocyte growth factor and mutations in the gene have been linked with development of hepatocellular carcinoma.

For the other down-regulated genes, the majority have not been associated to NAFLD or cirrhosis development, including PACSIN3. Most genes were found related to the development of different cancer types and metastasis, such as MAGI-A23, EIF3E, and SOX5, while PDSD3 was associated to NAFLD.

### 3.7. Gene ontology

We performed a gene ontology analysis to elucidate enhanced and downregulated biological pathways comparing our different study groups (Fig 3).

The most significantly downregulated biological processes for patients with NAFLD compared to healthy were related to cytoplasmatic translation and several ribosomal processes (Fig 3A). The most significant upregulated biological processes were related to wound healing and coagulation.

A

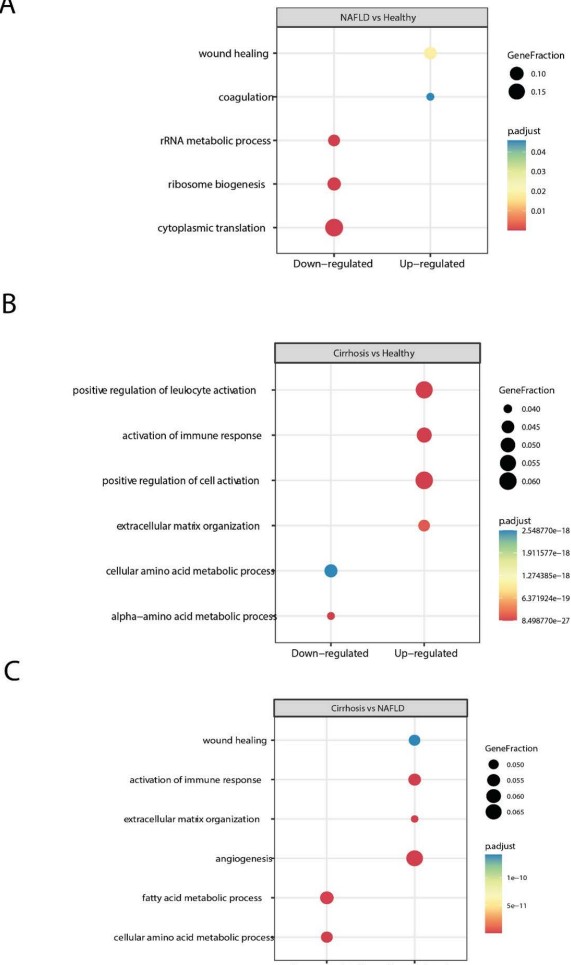

B

C

**Fig 3. Enrichment of Gene Ontology Biological Processes (GOBP).** (A) NAFLD vs healthy. (B) Cirrhosis vs healthy. (C) Cirrhosis vs NAFLD. Each dot reflects significant enrichment of GOBP that are either down- or upregulated in the respective groups. The size of the individual dot reflects the number of genes associated with the process whereas the colour represents the adjusted p-value in the comparison between groups.

Patients with cirrhosis, when comparing to healthy controls, showed a downregulation in processes related to metabolism, notably a downregulation in amino acid metabolism and cytoplasmatic translation (Fig 3B). The upregulated biological processes comparing cirrhosis with healthy were strongly related to immune response and extracellular matrix organization.

When comparing patients with cirrhosis and NAFLD (Fig 3C), we found a distinct downregulation of processes related to metabolism, especially amino acid and fatty acid metabolism. Among the most upregulated biological processes in patients with cirrhosis, we found various processes related to immune response, angiogenesis and wound healing, and extracellular matrix organization.

## 4. Discussion

This study provides an explorative data analysis of hepatic differential gene expression patterns in patients with NAFLD, cirrhosis, and healthy controls at both fasting and postprandial conditions. The latter has not, to our knowledge, been reported previously but our data suggest the liver transcriptome to be adaptive and although changes are subtle compared to group differences (cirrhosis versus healthy) we found >30 genes to be significantly altered postprandially. Some of these changes were blunted in patients with NAFLD and cirrhosis suggesting that the liver response to a meal may be impaired at a transcriptomic level in humans. The summed data are accessible through a freely browsable online tools https://weweralbrechtsenlab.shinyapps.io/PLS_Groups/ (differences between healthy, NAFLD, and cirrhosis) & https://weweralbrechtsenlab.shinyapps.io/PostprandialLiver/ (impact of a standardized meal on the liver transcriptome in healthy, NAFLD, and cirrhosis). The study has several limitations hereof in particular the limited sample size in the subgroup may cause heterogeneity and hence lack of statistical power. Age may affect mRNA levels of certain genes however based on a recent study the genes significantly affected in this study do not overlap with 'age dependent genes' [18].

Transcriptomic studies may be difficult to explore and compare. The number of genes, that may be described in a single paper, is naturally limited. Although more authors chose to make data publicly available using online data repositories, and more journals require data availability upon publication, these datasets are usually large and not browsable, there by limiting their use in future studies. We therefore developed a browsable online application to add a broader utility to our data. Here, researchers can easily browse their genes of interest to explore the differential expression in patients with NAFLD, cirrhosis, and healthy controls, both fasting and postprandial. Moreover, graphs and gene distributions can be easily downloaded and used in other publications or presentations. Another feature is the gene ontology biological process tab. Here, researchers can explore the gene ontology terms that are up- and downregulated between the groups and export the findings as dot plots or in an excel spreadsheet.

Most transcriptomic studies have focused on the differences between patients with NAFL and NASH. In this study, we included healthy controls and patients with cirrhosis providing a more complete spectrum of the hepatic transcriptome. It is one of the few studies including liver biopsies from healthy controls.

The transcriptional profiles differed substantially between groups, and we saw the greatest differences between patients with cirrhosis and healthy controls. Our NAFLD cohort consisted of patients with mostly simple steatosis, which resulted in a transcriptional profile that was closer to healthy controls than patients with cirrhosis. Still, we found a considerable number of differentially expressed genes when comparing NAFLD to healthy controls (553 genes). We found significantly upregulated genes in NAFLD compared to healthy that were related to metabolism, oxidative stress response, and steatosis. A few data highlights are discussed below.

Protein kinase c epsilon (PRKCE) is a protein coding gene with importance for hepatic insulin resistance, as recently investigated [19] was upregulated in NAFLD and cirrhosis. It is involved in different cellular processes such as apoptosis and insulin exocytosis and signalling. PRKCE was found to be upregulated in obese patients with NAFLD [20] compared with healthy, and high expression of PRCE in patients with NAFLD [21]. Perilipin1 (PLIN1) was found to be involved in the pathogenesis of hepatic steatosis [22] and is a modulator of adipocyte lipid metabolism with related pathways in glucose

and energy metabolism. Moreover, we found Isthmin1 (ISM1) and Serpin Family E Member 1 (SERPINE1) to be upregulated in NAFLD. ISM1 is a recently discovered adipokine that plays a role in glucose uptake regulation [23]. SERPINE1 encodes PAI-1 (plasminogen activator inhibitor 1) that plays an important role in hepatic lipid metabolism and was found to be a marker of the metabolic syndrome [24]. It acts as a proinflammatory marker that was found to be increased in adipocytes of obese patients with NAFLD but not obese patients without NAFLD [24]. Among the 10 most significantly upregulated genes in patients with NAFLD and cirrhosis was a gene related to oxidative stress response and autophagy, Tumor Protein P53 Inducible Nuclear Protein 1 (TP53INP1). A downregulation or inactivation of TP53INP3 has been linked to several cancer types including hepatocellular carcinoma. We found two novel candidate genes upregulated in NAFLD compared with healthy. The first one, T-Box Transcription Factor 15 (TBX15) was just recently identified as a novel "adipose master trans regulator of abdominal obesity genes" in a gene wise association study (GWAS) for polygenic risk for abdominal obesity and T2D in the UK biobank [25]. TBX15's role in NAFLD is to date not known. The second one, Zinc Finger And BTB Domain Containing 33 (ZBTB33) is a transcriptional regulator also known as "kaiso". Most previous studies on ZBTB33 are related to cancer, but one study in mice related ZBTB33 to PGC-1a linked to hepatic inflammation and apoptosis, which indicates a potential role of ZBTB33 in NAFLD in humans and should be further investigated [26].

Among the most significantly downregulated genes in patients with NAFLD and cirrhosis was the MET proto-oncogene, a receptor tyrosine kinase with Hepatocyte Growth Factor (HGF) as ligand that is important for cellular survival and cellular migration, as well as invasion. It may be important for organ regeneration and tissue remodelling, and MET alterations have been linked to human cancer. One study found MET signalling pathway to be protective from steatosis development and progression to steatohepatitis and fibrosis [27]. Another study found HGF to be protective for reactive oxygen species through inducing the glutathione-related protection system [28]. Interestingly, we found Growth Arrest Specific 5 (GAS5), a long noncoding RNA, to be downregulated in NAFLD. GAS5 was associated with protection against fibrosis [29]. One study found higher GAS5 plasma expression in patients with NAFLD and advanced fibrosis but not cirrhosis [30]. Lastly, two of the downregulated genes were related to cancer suppression and HCC. RBMS1 was identified as suppressor of colon cancer progression, and downregulation was negatively associated with patients' survival [31]. Fibronectin Type III Domain Containing 3B (FNDC3B) is a gene that enables RNA binding activity. It was found to promote cell migration and behaves like an oncogene, and a study found it downregulated in HCC in vitro and in vivo [32].

Apart from downregulated in NAFLD compared with healthy, MET was one of the most significantly downregulated gene in cirrhosis. Interestingly, the gene has not previously been associated with NAFLD or cirrhosis in humans. One study in mice, though, found that the deletion of MET led to the development of NASH [27], supporting a possible role of MET in the development of liver disease. A recent study on PDSD3 found a variant of PDSD3 (rs81518834) to be associated with lower liver fat content and thereby protective against NAFLD, NASH, and fibrosis development. PACSIN3 was found to be downregulated in response to IL-6 in mice [33] and in adipocytes, and an overexpression of PASIN3 increased glucose uptake by increasing GLUT1 [34]. The role in the liver has not been elucidated yet but should be subject to further research. Moreover, DNAJC12, a heat shock protein, was associated to ER stress, but has not been associated to NAFLD, cirrhosis, or metabolic disease yet [35]. SOX5, an important gene for embryonal development and chondrocyte differentiation, was associated to cancer development and metastasis, including HCC [36]. Although it was found to be upregulated in HCC and other cancers, we found SOX5 to be decreased in NAFLD and lowest in cirrhosis when compared to healthy controls. This might indicate an ambivalent role in liver disease when not related to HCC, and more research is needed to elucidate the role of SOX5 in metabolic liver disease.

### Gene ontology

The gene ontology biological process analyses mirrored the distinct hepatic transformations that occur in the course of liver disease. Processes related to cytoplasmatic translation were downregulated in patients with NAFLD, while genes related to wound healing and coagulation were upregulated. This might show a suboptimal liver function in patients with

NAFLD, where translational processes might be slowed down, and liver repair processes are upregulated to compensate for the liver injury associated to NAFLD and NASH. We found a distinct decrease in metabolism, as well as increase in pro-fibrotic processes in patients with NAFLD and cirrhosis. Patients with cirrhosis compared to NAFLD showed a decrease in amino acid and fatty acid metabolic processes, as well as decrease in amino acid metabolism compared to healthy controls. This highlights the decrease in metabolic agility in both NAFLD and especially cirrhosis, where the amount of functional liver tissue is decreased, and the normal metabolic function of the liver is impaired. The upregulated biological processes in cirrhosis were clearly related to fibrosis development and activation of the immune system, showing the chronic inflammatory state in patients with cirrhosis.

## Strengths and limitations

One strength of this study is that we cover the full spectrum of liver health and disease by including a control group of healthy young adults, patients with NAFLD, and patients with severe liver disease, cirrhosis. We used the golden standard, a liver biopsy, for final diagnosis of our participants. Another strength of the study is the use of unbiased, high-throughput state of the-art methods and advanced bioinformatics for data aggregation and analysis.

Our study also comes with some limitations. The sample size might be too small and too heterogenic to mirror the transcriptional profiles of the general population and the population of patients with liver disease. Moreover, the transcriptome is highly dynamic, and there can be large inter-personal variabilities. One might argue, though, that we found an abundance of differentially expressed genes previously described to be related to liver disease, indicating that our data might indeed give an accurate illustration of the hepatic transcriptional profiles in these patient groups. A limitation regarding the comparison between the fasting and the postprandial state might be the relatively short time of 60 minutes from intake of the meal to the liver biopsy. It is, however, also a strength of the study, and highlight the liver as a highly adaptive organ. It is possible that larger numerical changes can be detected after >1hours. We observed a relatively low number of DEG between fasting and postprandial, though, indicating that the study period was long enough to detect at least the early changes on the transcriptional level. Another limitation, that is natural for transcriptomic studies, is that the altered gene expression profiles might not translate into differential protein or metabolite patterns. Therefore, the use of multi-omics is needed to investigate differences in health and liver disease including the proteome and metabolome. Finally, given that our biopsies only represent a minor part of the whole liver we may potentially miss the identification of disease-specific regions and cell types [37]. Our data provided here may however be linked with some of the single cell-based atlases to uncover if the genes identified here represent cell specific features.

## Conclusion

This study investigated differential hepatic gene expressions between patients with NAFLD and cirrhosis as well as healthy controls. The results are easily accessible through a developed online browser to ensure broader usage of our data. Finally, our exploratory analyses of fasting and fed state in humans suggest that the hepatic transcriptome is much more adaptive than previously thought.

## Supporting information

**S1 Table. The 10 most up- and downregulated genes in NAFLD compared to healthy (fold change).**
(DOCX)

**S2 Table. Gene list of the most up-and downregulated genes in NAFLD vs cirrhosis and cirrhosis vs. NAFLD.**
(DOCX)

**S1 Fig. Study Design.**
(DOCX)

## Acknowledgments

The authors thank Stefan Stender (Department of Clinical Biochemistry, Copenhagen University Hospital – Rigshospitalet, University of Copenhagen, Denmark) for providing valuable input to study design, Christine Rasmussen (Department of Clinical Biochemistry, Bispebjerg Hospital, Copenhagen University Hospital Denmark) for excellent laboratory assistance, and technical staff at the department of Clinical Physiology for helpful assistance during the invasive procedures. We sincerely thank Department of Genomic Medicine, Rigshospitalet for aiding us to perform NGS.

## Author contributions

**Conceptualization:** Søren Møller, Lise Lotte Gluud, Nicolai J Wewer Albrechtsen.

**Data curation:** Nicolai J Wewer Albrechtsen.

**Formal analysis:** Christian D. Johansen, Lise Lotte Gluud, Nicolai J Wewer Albrechtsen.

**Funding acquisition:** Nicolai J Wewer Albrechtsen.

**Investigation:** Josephine Grandt, Christian D. Johansen, Anne-Sofie H. Jensen, Mikkel P. Werge, Elias B. Rashu, Andreas Møller, Anders E. Junker, Lise Hobolth, Christian Mortensen, Mogens Vyberg, Reza Rafiolsadat Serizawa, Søren Møller, Lise Lotte Gluud.

**Methodology:** Christian D. Johansen, Anne-Sofie H. Jensen, Lise Hobolth, Christian Mortensen, Mogens Vyberg, Reza Rafiolsadat Serizawa, Søren Møller, Lise Lotte Gluud.

**Project administration:** Josephine Grandt, Lise Lotte Gluud, Nicolai J Wewer Albrechtsen.

**Resources:** Christian D. Johansen, Lise Lotte Gluud, Nicolai J Wewer Albrechtsen.

**Supervision:** Søren Møller, Lise Lotte Gluud, Nicolai J Wewer Albrechtsen.

**Visualization:** Josephine Grandt, Christian D. Johansen.

**Writing – original draft:** Josephine Grandt, Christian D. Johansen, Nicolai J Wewer Albrechtsen.

**Writing – review & editing:** Anne-Sofie H. Jensen, Mikkel P. Werge, Elias B. Rashu, Andreas Møller, Anders E. Junker, Lise Hobolth, Christian Mortensen, Mogens Vyberg, Reza Rafiolsadat Serizawa, Søren Møller, Lise Lotte Gluud.

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
