## [Decision Letter · Decision Letter 0]

16 Dec 2024

PONE-D-24-26821The hepatic transcriptome is differentially regulated by a standardized meal in healthy individuals compared to patients with fatty liver diseasePLOS ONE Dear Dr. Wewer Albrechtsen,

Thank you for submitting your manuscript to PLOS ONE. After careful consideration, we feel that it has merit but does not fully meet PLOS ONE’s publication criteria as it currently stands. Therefore, we invite you to submit a revised version of the manuscript that addresses the points raised during the review process.

We look forward to receiving your revised manuscript.

Kind regards,

Mohamed El-Kassas

Academic Editor

PLOS ONE

Journal Requirements: When submitting your revision, we need you to address these additional requirements. 1. Please ensure that your manuscript meets PLOS ONE's style requirements, including those for file naming. The PLOS ONE style templates can be found at https://journals.plos.org/plosone/s/file?id=wjVg/PLOSOne_formatting_sample_main_body.pdf and https://journals.plos.org/plosone/s/file?id=ba62/PLOSOne_formatting_sample_title_authors_affiliations.pdf 2. We note that the grant information you provided in the ‘Funding Information’ and ‘Financial Disclosure’ sections do not match.  When you resubmit, please ensure that you provide the correct grant numbers for the awards you received for your study in the ‘Funding Information’ section. 3. Thank you for stating the following financial disclosure:  [The study and Josephine Grandt, Christian D. Johansen, Anne-Sofie H. Jensen, and Nicolai J. Wewer Albrechtsen were supported by Novo Nordisk Foundation Excellence Emerging Investigator Grant – Endocrinology and Metabolism (Application No. NNF19OC0055001), European Foundation for the Study of Diabetes Future Leader Award (NNF21SA0072746) and Independent Research Fund Denmark, Sapere Aude (1052-00003B). Novo Nordisk Foundation Center for Protein Research is supported financially by the Novo Nordisk Foundation (Grant agreement NNF14CC0001).].  Please state what role the funders took in the study.  If the funders had no role, please state: ""The funders had no role in study design, data collection and analysis, decision to publish, or preparation of the manuscript."" If this statement is not correct you must amend it as needed. Please include this amended Role of Funder statement in your cover letter; we will change the online submission form on your behalf. 4. Thank you for stating the following in the Acknowledgments Section of your manuscript: [The authors thank Stefan Stender (Department of Clinical Biochemistry, Copenhagen University Hospital - Rigshospitalet, University of Copenhagen, Denmark) for providing valuable input to study design, Christine Rasmussen (Department of Clinical Biochemistry, Bispebjerg Hospital, Copenhagen University Hospital Denmark) for excellent laboratory assistance, and technical staff at the department of Clinical Physiology for helpful assistance during the invasive procedures.We sincerely thank Department of Genomic Medicine, Rigshospitalet for aiding us to perform NGS.  The study and Josephine Grandt, Christian D. Johansen, Anne-Sofie H. Jensen, and Nicolai J. Wewer Albrechtsen were supported by Novo Nordisk Foundation Excellence Emerging Investigator Grant – Endocrinology and Metabolism (Application No. NNF19OC0055001), European Foundation for the Study of Diabetes Future Leader Award (NNF21SA0072746) and Independent Research Fund Denmark, Sapere Aude (1052-00003B). Novo Nordisk Foundation Center for Protein Research is supported financially by the Novo Nordisk Foundation (Grant agreement NNF14CC0001).]We note that you have provided funding information that is not currently declared in your Funding Statement. However, funding information should not appear in the Acknowledgments section or other areas of your manuscript. We will only publish funding information present in the Funding Statement section of the online submission form. Please remove any funding-related text from the manuscript and let us know how you would like to update your Funding Statement. Currently, your Funding Statement reads as follows:   [The study and Josephine Grandt, Christian D. Johansen, Anne-Sofie H. Jensen, and Nicolai J. Wewer Albrechtsen were supported by Novo Nordisk Foundation Excellence Emerging Investigator Grant – Endocrinology and Metabolism (Application No. NNF19OC0055001), European Foundation for the Study of Diabetes Future Leader Award (NNF21SA0072746) and Independent Research Fund Denmark, Sapere Aude (1052-00003B). Novo Nordisk Foundation Center for Protein Research is supported financially by the Novo Nordisk Foundation (Grant agreement NNF14CC0001).].  Please include your amended statements within your cover letter; we will change the online submission form on your behalf. 5. Please upload a copy of Figure 1, 2 and 3 to which you refer in your text on page 11, 13, 17. If the figure is no longer to be included as part of the submission please remove all reference to it within the text. 6. Please include captions for your Supporting Information files at the end of your manuscript, and update any in-text citations to match accordingly. Please see our Supporting Information guidelines for more information: http://journals.plos.org/plosone/s/supporting-information. 

Reviewers' comments:

Reviewer's Responses to Questions

**Comments to the Author**

1. Is the manuscript technically sound, and do the data support the conclusions?

Reviewer #1: Yes

Reviewer #2: Yes

2. Has the statistical analysis been performed appropriately and rigorously? 

Reviewer #1: I Don't Know

Reviewer #2: Yes

3. Have the authors made all data underlying the findings in their manuscript fully available?

Reviewer #1: Yes

Reviewer #2: Yes

4. Is the manuscript presented in an intelligible fashion and written in standard English?

Reviewer #1: Yes

Reviewer #2: Yes

5. Review Comments to the Author

Reviewer #1: The authors of a manuscript “PONE-D-24-26821: The hepatic transcriptome is differentially regulated by a standardized meal in healthy individuals compared to patients with fatty liver disease” present a study evaluating hepatic transcriptome at both fasting and postprandial states in patients with NAFLD, cirrhosis, and controls. They described huge number of differentially expressed genes (DEG) between healthy controls and patients with NAFLD and patients with cirrhosis. The authors also described that the hepatic transcriptome changed significantly during meal and the changes were blunted in patients with NAFLD and cirrhosis. They found an increase in pro-inflammatory and pro-fibrotic genes in NAFLD and cirrhosis. The main result is that the hepatic transcriptome is differentially regulated in healthy individuals compared to patients with NAFLD after meal.

Additionally, the authors developed a browsable resource enabling easy access to their data (two web-based applications).

Comments:

1. Healthy controls were markedly younger than patients (NAFLD, cirrhosis). Could the age influence the changes in transcriptome??

2. What was the aetiology of cirrhosis – NAFLD??

3. Did the authors measure HVPG during liver biopsy? If yes, did they observe any differences in transcriptome in relation to severity of portal hypertension??

4. The authors state that the presented study is one of the few studies including liver biopsies from healthy controls. Did the authors observe any complication during transjugular liver biopsy in healthy people?

This is an interesting study describing the differences in transcriptome in healthy controls, patients with NAFLD and patients with cirrhosis both fasting and after meal.

The added value is the creation of functional web applications for data viewing and analysis.

Reviewer #2: Summary

Grandt et al. investigated how food intake affects the hepatic transcriptome in patients with non-alcoholic fatty liver disease (NAFLD), cirrhosis, and healthy controls. Liver biopsies from these groups were analyzed to identify differentially expressed genes (DEGs) and gene ontology biological processes. Significant differences were found between the groups, with an increase in pro-inflammatory and pro-fibrotic genes in NAFLD and cirrhosis and a decrease in metabolic-related genes. In the presented experimental setting, food intake seemed to play a minor role in hepatic transcriptomes of healthy controls and diseased livers. The authors developed a web-based resources to make the data easily accessible.

General comment

Although the manuscript is structured and written comprehensibly, it lacks a clear presentation of scientific hypotheses. Clarifying the purpose of the study would strengthen the use of the online resource, a strong point of the study. Another positive aspect is the sampling via biopsies rather than post-mortem tissue collection, crucial for capturing rapid changes in liver metabolism and RNA expression. However, a biopsy only represents a small part of the whole organ, which could potentially lead to misleading results by missing disease-relevant regions and cell types. To mitigate this limitation, the authors could utilize publicly available (single-cell) datasets to put their findings in a broader context. Additionally, it would help to add histological assessments of the biopsies and validate some of the genes in situ or by PCR. If food intake does not drive significant transcriptional changes in the present experimental setting, this should also be reported and discussed.

Specific points

1. Please state clear hypothesis and idea: Clearly articulate the hypotheses and underlying ideas behind the study to strengthen the concept and manuscript. The phrases "Our aim was to investigate the hepatic transcriptome at both fasting and postprandial states in patients with..." or "Next-generation sequencing (NGS) provides a hypothesis-free approach with high specificity and sensitivity allowing insights into molecular mechanisms by identifying crucial genes..." are insufficient. Although the approach is conceptually "hypothesis-free," it is crucial to convey what insights or applications are anticipated from the data.

2. Clarify transcriptional changes: The statement "The hepatic transcriptome changed significantly during a standardized meal and these changes were blunted in patients with NAFLD and cirrhosis" needs clarification. Transcriptional changes between fasting and postprandial states appear minimal (PCA, number of DE genes). Please clarify this or present new data/analyses to substantiate the claim.

3. The number of DE genes appears high for some comparisons (e.g., cirrhosis samples). Ensure that potential batch effects were analyzed and corrected; if so, provide insights. Additionally, validating detected genes using orthogonal methods (in situ hybridization, immunohistochemistry, RT-PCR) would strengthen the biological message.

4. Provide histology of biopsies: Include histological assessments of the biopsies (“Part of the liver tissue was used for histological assessment”). This will help correlate histological findings with transcriptional changes.

5. Broader context: Integrating the present data into publicly available single-cell datasets would improve the study by providing context for transcriptional and potentially cellular changes in liver diseases. For example, see publications: DOI: 10.1016/j.xgen.2023.100301, DOI: 10.1016/j.isci.2021.103233)

6. Table 1: Please provide critical values/diagnostic thresholds, e.g. by using color green=normal vs red=above/lower than threshold. There might be regional differences in clinical standards/critical values.

7. A visual comparison of overlapping and non-overlapping DEGs would aid data presentation and evaluation. Consider using a Venn diagram.

8. PCA Loadings: Including PCA loadings could help describe differences between conditions. Consider adding the most informative loadings to the manuscript.

6. PLOS authors have the option to publish the peer review history of their article (what does this mean? ). If published, this will include your full peer review and any attached files.

**Do you want your identity to be public for this peer review?** For information about this choice, including consent withdrawal, please see our Privacy Policy .

Reviewer #1: **Yes: ** Radan Bruha

Reviewer #2: No

---

## [Decision Letter · Decision Letter 1]

12 Mar 2025

PONE-D-24-26821R1The hepatic transcriptome is differentially regulated by a standardized meal in healthy individuals compared to patients with fatty liver diseasePLOS ONE

Dear Dr. Wewer Albrechtsen,

Thank you for submitting your manuscript to PLOS ONE. After careful consideration, we feel that it has merit but does not fully meet PLOS ONE’s publication criteria as it currently stands. Therefore, we invite you to submit a revised version of the manuscript that addresses the points raised during the review process.

We look forward to receiving your revised manuscript.

Kind regards,

Mohamed El-Kassas

Academic Editor

PLOS ONE

Reviewers' comments:

Reviewer's Responses to Questions

**Comments to the Author**

1. If the authors have adequately addressed your comments raised in a previous round of review and you feel that this manuscript is now acceptable for publication, you may indicate that here to bypass the “Comments to the Author” section, enter your conflict of interest statement in the “Confidential to Editor” section, and submit your "Accept" recommendation.

Reviewer #1: All comments have been addressed

Reviewer #2: (No Response)

2. Is the manuscript technically sound, and do the data support the conclusions?

Reviewer #1: Yes

Reviewer #2: Partly

3. Has the statistical analysis been performed appropriately and rigorously? 

Reviewer #1: Yes

Reviewer #2: N/A

4. Have the authors made all data underlying the findings in their manuscript fully available?

Reviewer #1: Yes

Reviewer #2: Yes

5. Is the manuscript presented in an intelligible fashion and written in standard English?

Reviewer #1: Yes

Reviewer #2: Yes

6. Review Comments to the Author

Reviewer #1: All suggestions made to the authors of the paper have been adequately considered and explained.

Thanks.

Reviewer #2: Reviewer 2 Response

….

Author Response: Thank you for your time and effort and constructive inputs. We have done our best to revise the manuscript in line with your comments. We have not included publicly available dataset. All our data have been made publicly available and browsable for readers.

Reviewer 2 response: No, sorry. Most of the comments were ignored. Please check the uploaded revised manuscript - I only found four sentences in the revised version, which was added to the original submission.

1. Please state clear hypothesis and idea: Clearly articulate the hypotheses and underlying ideas behind the study to strengthen the concept and manuscript. The phrases "Our aim was to investigate the hepatic transcriptome at both fasting and postprandial states in patients with..." or "Next-generation sequencing (NGS) provides a hypothesis-free approach with high specificity and sensitivity allowing insights into molecular mechanisms by identifying crucial genes..." are insufficient. Although the approach is conceptually "hypothesis-free," it is crucial to convey what insights or applications are anticipated from the data.

Author Response: Revised accordingly. We hope this is sufficient and aligns with the comment by the reviewer.

Reviewer 2 response: Can not find the revised part for this issue – e.g. the abstract is identical from original submission.

2. Clarify transcriptional changes: The statement "The hepatic transcriptome changed significantly during a standardized meal and these changes were blunted in patients with NAFLD and cirrhosis" needs clarification. Transcriptional changes between fasting and postprandial states appear minimal (PCA, number of DE genes). Please clarify this or present new data/analyses to substantiate the claim.

Author Response: In table 3 and in the results section, we have provided details on

which genes are affected differently between disease state.

Reviewer 2 response: Point not addressed.

3. The number of DE genes appears high for some comparisons (e.g., cirrhosis samples). Ensure that potential batch effects were analyzed and corrected; if so, provide insights.

Additionally, validating detected genes using orthogonal methods (in situ hybridization,

immunohistochemistry, RT-PCR) would strengthen the biological message.

Author Response: Thank you for this point. The changes align with literature. We have

carefully performed the analysis in batches to circumvent such effects.

We have not performed additional verification as this does not align with any literature

on proteome analysis – or genomic for that sake. Sorry!

https://www.nature.com/articles/s41591-022-01850-y

Reviewer 2 response: Answer needs clarification: Where batch effects tested and detected?

“The changes align with literature” Are the results still new? If so, needs more validation for claims taken in the text.

“additional verification as this does not align with any literature” Very confusing answer. Are observed effects, e.g. number of DE genes, real?

4. Provide histology of biopsies: Include histological assessments of the biopsies (“Part of the liver tissue was used for histological assessment”). This will help correlate histological findings with transcriptional changes.

Author Response: The NAS scores are shown in table 2.

Reviewer 2 response: A NAS score is “only” an evaluation of the histology. Please provide images or reasons why this is not possible.

5. Broader context: Integrating the present data into publicly available single-cell datasets would improve the study by providing context for transcriptional and potentially cellular changes in liver diseases. For example, see publications: DOI: 10.1016/j.xgen.2023.100301, DOI: 10.1016/j.isci.2021.103233)

Author Response: We have included the two references in the revised manuscript.

Reviewer 2 response: Reference were not added and were not used to validate or extend the analysis.

6. Table 1: Please provide critical values/diagnostic thresholds, e.g. by using color

green=normal vs red=above/lower than threshold. There might be regional differences in

clinical standards/critical values.

Author Response: Good point. However, the references are internationally used. If we

are allowed by the editor, we can provide color coding.

Reviewer 2 response: Labeling can be done by symbols or expressed by using text (low, normal, high)

7. A visual comparison of overlapping and non-overlapping DEGs would aid data presentation and evaluation. Consider using a Venn diagram.

Author Response: The DEG are shown in table 3.

Reviewer 2 response: Comment was ignored.

8. PCA Loadings: Including PCA loadings could help describe differences between conditions. Consider adding the most informative loadings to the manuscript.

Author Response: Thank you. We have two panels one showing differences between

conditions and one between fasting an postprandial. We hope this is in line with your

suggestions.

Reviewer 2 response: not addressed

7. PLOS authors have the option to publish the peer review history of their article (what does this mean? ). If published, this will include your full peer review and any attached files.

**Do you want your identity to be public for this peer review?** For information about this choice, including consent withdrawal, please see our Privacy Policy .

Reviewer #1: No

Reviewer #2: No

---

## [Editor Report · Decision Letter 2]

25 Apr 2025

The hepatic transcriptome is differentially regulated by a standardized meal in healthy individuals compared to patients with fatty liver disease

PONE-D-24-26821R2

Dear Dr. Albrechtsen

We’re pleased to inform you that your manuscript has been judged scientifically suitable for publication and will be formally accepted for publication once it meets all outstanding technical requirements.

Kind regards,

Mohamed El-Kassas

Academic Editor

PLOS ONE
---

## [Editor Report · Acceptance letter]

PONE-D-24-26821R2

PLOS ONE

Dear Dr. Wewer Albrechtsen,

I'm pleased to inform you that your manuscript has been deemed suitable for publication in PLOS ONE. Congratulations! Your manuscript is now being handed over to our production team.

Kind regards,

on behalf of

Professor Mohamed El-Kassas

Academic Editor

PLOS ONE